# UniAP: Unifying Inter- and Intra-Layer Automatic Parallelism by Mixed Integer Quadratic Programming

## Abstract

Deep learning models have demonstrated impressive performance in various domains. However, the prolonged training time of these models remains a critical problem. Manually designed parallel training strategies could enhance efficiency but require considerable time and deliver little flexibility. Hence, *automatic parallelism* is proposed to automate the parallel strategy searching process. Even so, existing approaches suffer from sub-optimal strategy space because they treat automatic parallelism as two independent processes, namely inter- and intra-layer parallelism. To address this issue, we propose UniAP, which utilizes mixed integer quadratic programming to unify inter- and intra-layer automatic parallelism. To the best of our knowledge, UniAP is the first work to optimize these two parallelism dimensions jointly for a globally optimal strategy. The experimental results show that UniAP outperforms state-of-the-art methods by up to $1.7\times$ in throughput and reduces strategy searching time by up to $107\times$ across four Transformer-like models.

## 1 Introduction

Deep learning models have been widely used in many applications. For example, BERT (Devlin et al., 2019), GPT-3 (Brown et al., 2020), and T5 (Raffel et al., 2020) achieved state-of-the-art (SOTA) results on different natural language processing (NLP) tasks. For computer vision (CV), Transformer-like models such as ViT (Dosovitskiy et al., 2021) and Swin Transformer (Liu et al., 2021) deliver excellent accuracy performance upon multiple tasks.

At the same time, training deep learning models has been a critical problem troubling the community due to the long training time, especially for those large models with billions of parameters (Brown et al., 2020). In order to enhance the training efficiency, researchers propose some manually designed parallel training strategies (Narayanan et al., 2021b; Shazeer et al., 2018; Xu et al., 2021). However, selecting, tuning, and combining these strategies require extensive domain knowledge in deep learning models and hardware environments. With the increasing diversity of modern hardware architectures (Flynn, 1966; 1972) and the rapid development of deep learning models, these manually designed approaches are bringing heavier burdens to developers. Hence, *automatic parallelism* is introduced to automate the parallel strategy searching for training models.

There are two main categories of parallelism in deep learning models: inter-layer parallelism (Huang et al., 2019; Narayanan et al., 2019; 2021a; Fan et al., 2021; Li & Hoefler, 2021; Lepikhin et al., 2021; Du et al., 2022; Fedus et al., 2022) and intra-layer parallelism (Li et al., 2020; Narayanan et al., 2021b; Rasley et al., 2020; FairScale authors, 2021). Inter-layer parallelism partitions the model into disjoint sets on different devices without slicing tensors. Alternatively, intra-layer parallelism partitions tensors in a layer along one or more axes and distributes them across different devices.

Current automatic parallelism techniques focus on optimizing strategies within these two categories. However, they treat these two categories separately. Some methods (Zhao et al., 2022; Jia et al., 2018; Cai et al., 2022; Wang et al., 2019; Jia et al., 2019; Schaarschmidt et al., 2021; Liu et al., 2023) overlook potential opportunities for inter- or intra-layer parallelism, the others optimize inter- and intra-layer parallelism hierarchically (Narayanan et al., 2019; Fan et al., 2021; He et al., 2021; Tarnawski et al., 2020; 2021; Zheng et al., 2022). As a result, current automatic parallelism techniques

often fail to achieve the global optima and instead become trapped in local optima. Therefore, a unified approach to jointly optimize inter- and intra-layer parallelism is needed to enhance the effectiveness of automatic parallelism.

This paper aims to find the optimal parallelism strategy while simultaneously considering inter- and intra-layer parallelism. It enables us to search in a more extensive strategy space where the globally optimal solution lurk. However, optimizing inter- and intra-layer parallelism jointly brings us two challenges. Firstly, to apply a joint optimization process on the inter- and intra-layer automatic parallelism, we should not formalize them with separate formulations as prior works. Therefore, how can we express these parallelism strategies in a unified formulation? Secondly, previous methods take a long time to obtain the solution with a limited strategy space. Therefore, how can we ensure that the best solution can be obtained in a reasonable time while expanding the strategy space?

To solve the above challenges, we propose UniAP. For the first challenge, UniAP adopts the mixed integer quadratic programming (MIQP) (Lazimy, 1982) to search for the globally optimal parallel strategy automatically. It unifies the inter- and intra-layer automatic parallelism in a single MIQP formulation. For the second challenge, our complexity analysis and experimental results show that UniAP can obtain the globally optimal solution in a significantly shorter time.

The contributions of this paper are summarized as follows:

- We propose UniAP, the first framework to optimize inter- and intra-layer automatic parallelism jointly with a unified formulation.
- UniAP demonstrates its scalability in terms of training throughput of optimal parallel strategies and strategy searching time.
- The experimental results show that UniAP speeds up model training on four Transformer-like models by up to $1.7\times$ and reduces the strategy searching time by up to $107\times$, compared with the SOTA method.

## 2 BACKGROUND

### 2.1 INTER- AND INTRA-LAYER PARALLELISM

In general, there exist two main categories of parallelism strategies for deep learning models: inter- and intra-layer parallelism. If we want to divide them further, inter-layer parallelism mainly includes pipeline parallelism (PP) in our context. Meanwhile, intra-layer parallelism mainly includes data parallelism (DP), tensor parallelism (TP), and fully sharded data parallelism (FSDP). Most manual and automatic parallelism approaches search for the optimal strategy within these dimensions.

### 2.2 MANUAL PARALLELISM

Manual parallelism refers to parallel computing strategies designed and optimized by human experts. Representative methods include Megatron-LM (Narayanan et al., 2021b), Mesh-TensorFlow (Shazeer et al., 2018), and GSPMD (Xu et al., 2021). Megatron-LM is a high-performance computing library for parallel Transformer training. It exhibits superior efficiency in both computing and scaling on clusters. Mesh-TensorFlow and GSPMD require users to annotate the desired intra-layer parallel computing mode. Such methods rely on expert design and manual tuning, challenging their automatic application to other models.

### 2.3 AUTOMATIC PARALLELISM

**Inter- or intra-layer-only automatic parallelism** For inter-layer-only automatic parallelism, GPipe (Huang et al., 2019) and vPipe (Zhao et al., 2022) employ a balanced partition algorithm and a dynamic layer partitioning middleware to partition pipelines, respectively. The parallel strategies they generate could be more optimal because both algorithms are greedy. For intra-layer-only automatic parallelism, OptCNN (Jia et al., 2018), TensorOpt (Cai et al., 2022), and Tofu (Wang et al., 2019) employ dynamic programming methods to solve DP and TP together. Meanwhile, FlexFlow (Jia et al., 2019) and Automap (Schaarschmidt et al., 2021) use a Monte Carlo approach to find the parallel execution plan. Colossal-Auto (Liu et al., 2023) utilizes integer programming techniques to

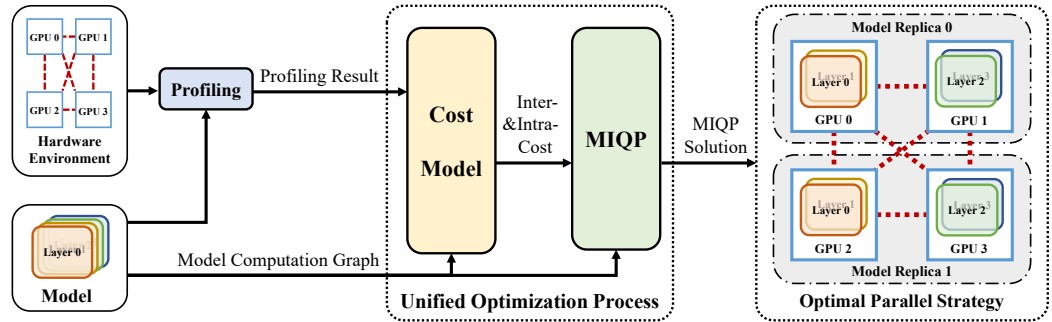

Figure 1: Overview of UniAP.

generate strategies for intra-layer parallelism. These methods explore a more limited strategy space for automatic parallelism and do not produce a globally optimal solution.

**Inter- and intra-layer automatic parallelism**   Auto-MAP (Wang et al., 2020) presents a Deep Q-Network (DQN) for DP-only, TP-only, and PP-only strategy searching, which requires relatively high model training time. PipeDream (Narayanan et al., 2019), DAPPLE (Fan et al., 2021), and PipeTransformer (He et al., 2021) use pure dynamic programming to determine optimal strategies for both DP and PP. DNN-partitioning (Tarnawski et al., 2020) adopts integer and dynamic programming to explore DP and PP strategies. All of these approaches neglect potential optimization opportunities in TP. Piper (Tarnawski et al., 2021) and Alpa (Zheng et al., 2022) adopt a hierarchical approach to automatic parallelism, considering DP, TP, and PP. The main difference is that Piper searches for strategies in layer granularity, while Alpa searches for operator granularity. This perspective produces locally near-optimal solutions rather than globally optimal ones. Galvatron (Miao et al., 2022) uses pure dynamic programming to determine DP, TP, and FSDP strategies on a single pipeline stage. As for PP, it partitions stages and determines micro-batch size using naive greedy algorithms. Compared with them, UniAP holds the most extensive search space for PP, DP, TP, and FSDP.

## 3 METHOD

### 3.1 OVERVIEW

As shown in Figure 1, UniAP initially profiles runtime information for both the deep learning model and the user's hardware. After that, UniAP estimates inter- and intra-layer costs given the computation graph and profiling results with its cost models. The estimated costs, along with the computation graph, are then transformed into a MIQP problem. The objective function of the MIQP is to maximize the training throughput, or in other words, to minimize the time-per-iteration (TPI). During its optimization process, the off-the-shelf solver will guarantee optimality. By iteratively applying the cost model and solver with different parameters, UniAP determines the globally minimal TPI and its corresponding parallel strategies. We name this process the Unified Optimization Process (UOP). Finally, UniAP interprets the solution into the execution plan for the designated model.

### 3.2 STRATEGY SPACE

**Pipeline parallelism**   In PP, each worker holds a disjoint set of model layers. Adjacent layers on different workers need to transfer activations in the forward propagation (FP) step and gradients in the backward propagation (BP) step. UniAP focuses on synchronous PP, which performs weight updating in each stage at the end of each iteration.

**Data parallelism**   In DP, each worker holds a replica of the model and uniformly partitioned training samples. In each iteration during training, each worker computes gradients and synchronizes them with the other workers using an all-reduce collective communication (CC). All workers will observe the same model parameters after the synchronization step.

**Tensor parallelism** In TP, each worker holds a partition of the model and a replica of training samples. In each iteration, each worker computes its local output in FP and its local gradients in BP. If the tensor is sliced uniformly, all workers will perform the same all-reduce CC in FP and BP steps.

**Fully sharded data parallelism** The FSDP approach involves partitioning optimizer states, parameters, and gradients of the model into separate workers. During the FP and BP step of each iteration, FSDP performs an all-gather CC to obtain the complete parameters for the relevant layer, respectively. Following the computation of gradients, FSDP conducts a reduce-scatter CC to distribute the global gradients among the workers.

### 3.3 PROFILING AND COST MODEL

UniAP collects runtime information for the hardware environment and the specific model during profiling. Regarding the hardware environment, UniAP assesses the efficiency of all-reduce and point-to-point (P2P) communication for various sets of devices. For instance, when profiling a node equipped with 4 GPUs, UniAP measures the all-reduce efficiency for various combinations of DP, TP, and FSDP across these GPUs. Additionally, UniAP ranks these GPUs from 0 to 3 and evaluates the speed of P2P for two pipeline options: $(0 \rightarrow 2$ and $1 \rightarrow 3)$ and $(0 \rightarrow 1, 1 \rightarrow 2$ and $2 \rightarrow 3)$. Furthermore, UniAP estimates the computation-communication overlap coefficient (CCOC), a metric previously explored by Miao et al. (2022) and Rashidi et al. (2021). Regarding the specific model, UniAP analyzes the forward computation time per sample for different types of hidden layers during the profiling process.

UniAP employs two primary cost models, namely the time cost model and the memory cost model. To model the computation time, UniAP first multiplies the batch size and the forward computation time per sample obtained from profiling to estimate the forward computation time. For Transformer-like models that mainly consist of the MatMul operator, the computation time in the BP stages is roughly twice that of the FP stages (Narayanan et al., 2021b; Li & Hoefler, 2021; Miao et al., 2022). Additionally, UniAP estimates the communication time by dividing the size of transmitting tensors by the profiled communication efficiency for different communication primitives. To account for overlapping, UniAP multiplies the profiled CCOC by the overlapping interval of computation and communication. To model the communication time between pipeline stages, UniAP calculates the cross-stage cost between consecutive stages as the summation of P2P costs.

In addition to the time cost model, UniAP estimates the memory consumption in GPUs by multiplying the tensor's shape and data type for the memory cost model. Furthermore, the memory cost model takes the context memory and activation memory into account. Overall, the cost models employed by UniAP strike a balance between complexity and accuracy.

### 3.4 MIXED INTEGER QUADRATIC PROGRAMMING

This section describes our MIQP expression in terms of a formulation-oriented approach.

To begin with, it is necessary to set up a function to model our objective, which is minimizing TPI. Currently, we choose GPipe as our PP algorithm for simplicity while preserving generality. Figure 2 depicts a typical GPipe scheduling process that incurs a non-negligible communication overhead. The time required for applying gradients at the end of each iteration is excluded, as it is both dependent on the optimizer and is negligible in comparison to the overall time spent on FP and BP.

We denote the cost for computation stages as $\mathbb{P} = \{p_1, p_2, \ldots, p_{deg}\}$ and the cost for communication stages as $\mathbb{O} = \{o_1, o_2, \ldots, o_{deg-1}\}$. Here, $deg$ represents the number of computation stages, which corresponds to the degree of PP. $fp_i$ and $bp_i$ means forward and backward computation time for computation stage $i$, respectively. Meanwhile, $fo_j$ and $bo_j$ means forward and backward communication time for communication stage $j$, respectively. Hence, we have $p_i = fp_i + bp_i$ and $o_j = fo_j + bo_j$.

In a GPipe-style pipeline, we denote $c$ as chunk size, which is equivalent to the number of micro-batches. As visualized in Figure 2, a mini-batch is uniformly split into three chunks and the total TPI is determined by the latency of all computation stages and communication stages and the latency of the slowest stage. We further denote $tpi$ as TPI in GPipe. Given that a stage with a higher FP computation cost leads to a higher BP computation cost with high probability, we could summarize

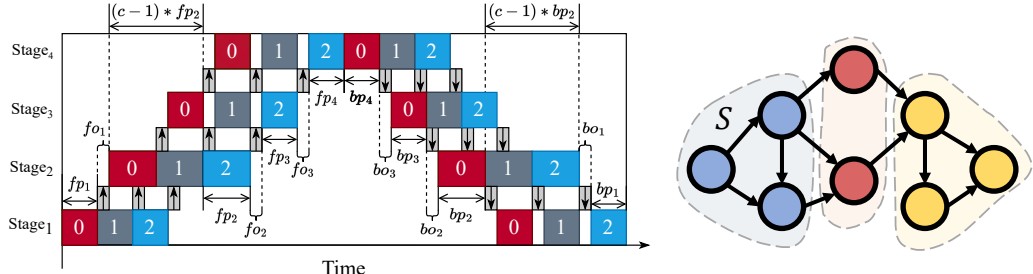

Figure 2: Cost decomposition of a GPipe-style PP.  Figure 3: A contiguous set $S$.

the TPI of GPipe-style pipeline as follows:

$$tpi = \sum_{i=1}^{deg} p_i + \sum_{j=1}^{deg-1} o_j + (c-1)\max\left(\mathbb{P} \cup \mathbb{O}\right). \tag{1}$$

Subsequently, we contemplate which aspects should be considered in the constraints of the MIQP expression. We list our main thoughts below:

1. In order to determine the total overhead for a single computation stage $i$, it is necessary to aggregate all computation and communication costs associated with that stage and assign them to $p_i$;

2. To calculate the total overhead for a single communication stage $j$, we should sum the P2P costs incurred between consecutive stages and assign them to $o_j$;

3. We should guarantee that no GPUs will encounter out-of-memory (OOM) exceptions;

4. The computation graph of the model must be partitioned into contiguous subgraphs to prevent disordered assignment to different pipeline stages.

Among them, the last point might be the most ambiguous one. We further explain it here. Typically, we can represent a deep learning model as a directed acyclic graph (DAG), namely $\mathcal{G}(V, E)$. Here, $V$ represents all layers in the model, while $E$ represents all edge connections between these layers. We borrow the definition of contiguity from Tarnawski et al. (2020; 2021).

**Definition 1.** *A set $S \subseteq V$ is contiguous if there do not exist nodes $u \in S$, $v \in V \setminus S$, and $w \in S$ such that $v$ is reachable from $u$ and $w$ is reachable from $v$.*

As Figure 3 illustrates, we cannot find any reachable node pairs $\langle u, v \rangle$ and $\langle v, w \rangle$ where $u, w \in S$ and $v \in V \setminus S$. Therefore, the set $S$ is considered contiguous. In our scenario, our model will not be assigned to different pipeline stages in a disordered fashion if we make sure that all subgraphs on each computation stage are contiguous.

Based on the above considerations, the MIQP formulation can be formalized as follows:

$$\min \quad tpi = \sum_{i=1}^{deg} p_i + \sum_{j=1}^{deg-1} o_j + (c-1)\max\left(\mathbb{P} \cup \mathbb{O}\right), \tag{MIQP}$$

$$\text{s.t.} \quad \sum_{u \in V} P_{ui}S_u^\mathsf{T}A_u + \sum_{\langle u,v \rangle \in E} P_{ui}P_{vi}(S_u^\mathsf{T}R_{uv}S_v) = p_i, \quad \forall i \in \{1,\ldots,deg\}, \tag{2}$$

$$\sum_{\langle u,v \rangle \in E} P_{uj}P_{v(j+1)}(S_u^\mathsf{T}R'_{uv}S_v) = o_j, \quad \forall j \in \{1,\ldots,deg-1\}, \tag{3}$$

$$\sum_{u \in V} P_{ui}S_u^\mathsf{T}M_u \leqslant m, \quad \forall i \in \{1,\ldots,deg\}, \tag{4}$$

$$V_i = \{\forall u \in V : P_{ui} = 1\} \text{ is contiguous}, \quad \forall i \in \{1,\ldots,deg\}, \tag{5}$$

$$\sum_{i=1}^{deg} P_{ui} = 1, \quad \forall u \in V, \tag{6}$$

$$\sum_{u \in V} P_{ui} \geqslant 1, \qquad\qquad \forall i \in \{1, \ldots, deg\}, \qquad\qquad (7)$$

$$\sum_{k=1}^{|g_u|} S_{uk} = 1, \qquad\qquad \forall u \in V, \qquad\qquad (8)$$

$$P_{ui} \in \{0, 1\}, \qquad\qquad \forall u \in V,\ i \in \{1, \ldots, deg\}, \qquad (9)$$

$$S_{uk} \in \{0, 1\}, \qquad\qquad \forall u \in V,\ k \in \{1, \ldots, |g_u|\}. \qquad (10)$$

For a given layer $u \in V$, we utilize the following notations: $g_u$ represents its set of intra-layer parallel strategies, $A_{uk}$ denotes the $k$-th intra-layer execution cost obtained from our time cost model, and $M_{uk}$ denotes the $k$-th intra-layer memory cost on a single device obtained from our memory cost model. Additionally, we use $S_{uk}$ as a 0-1 variable indicating whether the $k$-th parallel strategy is selected for the layer $u$, and $P_{ui}$ as a 0-1 variable indicating whether layer $u$ is to be placed on the $i$-th computation stage. Each edge $\langle u, v \rangle \in E$ is assigned a resharding cost denoted by $R_{uv}$ if the vertices are located within the same pipeline stage. Alternatively, if the vertices are located across consecutive stages, the resharding cost between them is denoted by $R'_{uv}$. These two resharding costs are constant matrices derived from our time cost model.

We explain the constraints as follows:

1. Equation 2 encodes the summation of intra-stage computation and communication costs as $p_i$. The first term of the polynomial represents the cost of choosing some particular intra-layer strategies for layers placed in stage $i$. The second term represents total resharding costs in stage $i$. Thus, this constraint term formalizes the first point of our thoughts.

2. Equation 3 encodes the inter-stage communication cost between consecutive computation stages as $o_j$. This term formalizes the second point of our thoughts.

3. Equation 4 formalizes the third point of our thoughts with a memory limit of $m$ for each device. In the case of homogeneous computing devices, the value of $m$ remains constant throughout all stages. However, the value of $m$ varies in the case of heterogeneous devices.

4. Equation 5 represents the last point of our thoughts. It is worth noting that we can formulate this constraint as a set of linear constraints as follows. Intuitively, $Z_{vi} = 1$ if there exists a node $w \in S$ reachable from $v$. Otherwise, $Z_{vi} = 0$. Please refer to Appendix A for proofs.

$$Z_{vi} \geqslant P_{vi}, \qquad \forall v \in V,\ \forall i \in \{1, 2, \ldots, deg\}, \qquad\qquad (11)$$

$$Z_{vi} \leqslant Z_{ui}, \qquad \forall u, v \in V,\ \forall \langle u, v \rangle \in E,\ \forall i \in \{1, 2, \ldots, deg\}, \qquad (12)$$

$$Z_{vi} \leqslant P_{vi} - P_{ui} + 1, \quad \forall u, v \in V,\ \forall \langle u, v \rangle \in E,\ \forall i \in \{1, 2, \ldots, deg\}. \qquad (13)$$

5. Equation 6, equation 7 and equation 9 represent that all layers should be placed on exactly one pipeline stage and at least one layer should be placed on each pipeline stages.

6. Equation 8 and equation 10 represent that each layer should choose exactly one strategy.

UniAP gets the minimum TPI and all its corresponding parallel strategies by solving the above MIQP expression using an off-the-shelf solver.

## 3.5 UNIFIED OPTIMIZATION PROCESS

In this section, we propose our design for UOP in UniAP. In short, UOP is mainly responsible for invoking the cost model and MIQP algorithms based on the profiling results and the computation graph. It eventually returns the globally optimal strategy and the corresponding TPI.

First, UOP considers pure intra-layer parallelism. Several works (Zheng et al., 2022; Liu et al., 2023) have adopted quadratic integer programming (QIP) to solve it and achieved promising results. UniAP provides a QIP formulation for intra-layer-only parallelism in Appendix B.

Then, UOP enumerates the pipeline degree $deg$ for the PP strategy from 2 to $n$ exponentially. For each $deg$, UOP enumerates the chunk size from 2 to mini-batch size one by one and selects those divisible by the mini-batch size to ensure load balancing across micro-batches. Here, we assume the number of devices is a power of 2 and these devices are homogeneous. This assumption is made to provide a more intuitive explanation of the overall process and how load balancing is achieved.

However, UOP is not limited to the specific case and can be extended to other scenarios with simple modifications as well.

For each candidate pipeline degree $deg$ and chunk size $c$, UOP calculates the cost and waits for the MIQP solver to return the optimal cost and parallelism strategy under the current configuration. Eventually, UOP will return the minimum cost $cost_{min}$ and its corresponding pipeline degree $deg_{min}$, chunk size $c_{min}$, layer placement $P_{min}$, and intra-layer strategies $S_{min}$.

Algorithm 1 summarizes this process. In the algorithm, we further denote intra-layer cost as $A$, inter-layer cost as $R$, cross-stage cost as $R'$, and memory cost as $M$. The `CalculateCost` process calculates these four costs based on the cost model described in Section 3.3. Let $|V|$, $|g|$, and $n$ denote the number of layers, parallel strategies, and GPUs, respectively. The time complexity of the UOP algorithm is $\mathcal{O}(|V||g|\log(n))$. For a more detailed analysis, please refer to Appendix C.

---

**Algorithm 1** Unified Optimization Process

---

**Input:** Profiling results $PR$, strategy dictionary $SD$, mini-batch size $B$, computation graph $\mathcal{G}$, and the number of GPUs $n$.
**Output:** Optimal cost $cost_{min}$, pipeline degree $deg_{min}$, chunk size $c_{min}$, layer placement $P_{min}$, and intra-layer strategy $S_{min}$
$deg_{min}, c_{min} = 1, B$
$A, R, \_, M = \texttt{CalculateCost}(PR, SD[1], \mathcal{G}, B)$;
$cost_{min}, P_{min}, S_{min} = \texttt{QIP}(A, R, M)$;
**for** $deg$ in $\{2, 4, \ldots, n\}$ **do**
  **for** $c = 2$ **to** $B$ **and** $c \mid B$ **do**
    Micro-batch size $b = B/c$;
    $A, R, R', M = \texttt{CalculateCost}(PR, SD[deg], \mathcal{G}, b)$;
    $cost, P, S = \texttt{MIQP}(A, R, R', M, deg, c)$;
    **if** $cost < cost_{min}$ **then**
      $cost_{min}, deg_{min}, c_{min}, P_{min}, S_{min} = cost, deg, c, P, S$;
    **end if**
  **end for**
**end for**

---

## 4 Experiment

UniAP utilizes the Gurobi Optimizer (Gurobi Optimization, LLC, 2023) to solve the MIQP problem. We conduct experiments on three kinds of environments. *EnvA* refers to a node with 1 Xeon 6248 CPU, 8 V100-SXM2 32GB GPUs, and 472GB memory. *EnvB* refers to two nodes interconnected with 10Gbps networks, where each node has 2 Xeon E5-2620 v4 CPUs, 4 TITAN Xp 12GB GPUs, and 125GB memory. *EnvC* has four nodes, each with the same configuration as that in *EnvB*.

We evaluate UniAP with four Transformer-like models, BERT-Huge (Devlin et al., 2019), T5-Large (Raffel et al., 2020), ViT-Huge (Dosovitskiy et al., 2021), and Swin-Huge (Liu et al., 2021) with different mini-batch sizes. Overall, we follow the common practice of training these Transformer-like models. However, to ensure fairness, we disable techniques orthogonal to parallel strategies, such as mixed precision training (Micikevicius et al., 2018) and activation checkpointing (Chen et al., 2016).

Our experimental evaluation focuses on two primary metrics: training throughput and strategy searching time. The former metric is computed by averaging throughput from the 10th to the 60th iteration of training, while the latter is determined by measuring the time of the UOP. Further elaborations are available in Appendix D.

### 4.1 Training throughput and strategy searching time

We compare the throughput of the optimal parallel strategy and UniAP's strategy searching time with the baseline approach. For experiments conducted on *EnvA*, we select 32, 16, 128, and 128 as the mini-batch size for BERT, T5, ViT, and Swin, respectively. As for experiments conducted on *EnvB*, we set the mini-batch size as 16, 8, 64, and 32 for these four models.

Table 1: Training throughput and strategy searching time on four Transformer-like models. The number following the model's name represents the number of hidden layers in the corresponding model. `MEM×` indicates out-of-memory (OOM) exceptions during strategy searching, while `CUDA×` indicates CUDA OOM exceptions during model training. Due to the absence of an official JAX implementation for Swin-Huge in HuggingFace Transformers (v4.29), experiments involving Swin-Huge for Alpa have been excluded and denoted as `N/A`. Additionally, the minimum and maximum training throughput speedup is calculated by dividing the average throughput of UniAP by the maximum and minumum average throughput of Galvatron and Alpa, repectively. Similarly, the minimum and maximum strategy searching time speedup is obtained by dividing the minimum and maximum average search time of Galvatron and Alpa by the average search time of UniAP, repectively.

| Env. | Model | Training throughput (samples/s) | | | Minimum speedup | Maximum speedup |
|------|-------|--------|------|-------|-----|-----|
| | | Galvatron | Alpa | UniAP | | |
| *EnvA* | BERT-Huge-32 | **33.46 ± 0.28** | 31.56 ± 0.04 | **33.46 ± 0.28** | 1.00 | 1.06 |
| | T5-Large-48 | **23.29 ± 0.04** | `MEM×` | **23.29 ± 0.04** | 1.00 | 1.00 |
| | ViT-Huge-32 | **109.51 ± 0.07** | 97.66 ± 1.42 | **109.51 ± 0.07** | 1.00 | 1.12 |
| | Swin-Huge-48 | `CUDA×` | `N/A` | **67.96 ± 0.12** | `N/A` | `N/A` |
| *EnvB* | BERT-Huge-32 | 6.27 ± 0.17 | 8.95 ± 0.06 | **10.77 ± 0.13** | 1.20 | 1.71 |
| | T5-Large-32 | **8.06 ± 0.06** | `MEM×` | 7.98 ± 0.05 | 0.99 | 0.99 |
| | ViT-Huge-32 | 32.20 ± 0.17 | 38.74 ± 0.20 | **45.58 ± 0.54** | 1.18 | 1.41 |
| | Swin-Huge-48 | 13.90 ± 0.17 | `N/A` | **19.08 ± 0.10** | 1.37 | 1.37 |

| Env. | Model | Strategy searching time (min.) | | | Minimum speedup | Maximum speedup |
|------|-------|--------|------|-------|-----|-----|
| | | Galvatron | Alpa | UniAP | | |
| *EnvA* | BERT-Huge-32 | 6.44 ± 0.588 | > 40 | **0.37 ± 0.002** | 17.29 | > 107.41 |
| | T5-Large-48 | 12.41 ± 0.122 | `MEM×` | **0.89 ± 0.007** | 13.98 | 13.98 |
| | ViT-Huge-32 | 6.29 ± 0.464 | > 40 | **0.57 ± 0.009** | 10.95 | > 69.60 |
| | Swin-Huge-48 | 11.88 ± 0.666 | `N/A` | **2.16 ± 0.004** | 5.49 | 5.49 |
| *EnvB* | BERT-Huge-32 | 2.04 ± 0.010 | > 40 | **1.51 ± 0.005** | 1.34 | > 26.32 |
| | T5-Large-32 | 2.64 ± 0.110 | `MEM×` | **0.91 ± 0.005** | 2.90 | 2.90 |
| | ViT-Huge-32 | 2.37 ± 0.180 | > 40 | **1.11 ± 0.011** | 2.14 | > 36.01 |
| | Swin-Huge-48 | 4.29 ± 0.320 | `N/A` | **2.29 ± 0.010** | 1.87 | 1.87 |

We have selected Galvatron (Miao et al., 2022) and Alpa (Zheng et al., 2022) as our baseline due to their recognition as the SOTA methods. Specifically, Galvatron has surpassed existing methods, including PyTorch DDP (Li et al., 2020), Megatron-LM (Narayanan et al., 2021b), FSDP (FairScale authors, 2021; Rajbhandari et al., 2020), GPipe (Huang et al., 2019), and DeepSpeed 3D (Microsoft, 2021) in terms of training throughput, as reported in its original publication (Miao et al., 2022). Additionally, Alpa utilizes the Just-In-Time (JIT) compilation feature in JAX and outperforms Megatron-LM and DeepSpeed.

Table 1 presents the training throughput and strategy search time on *EnvA* and *EnvB*. On *EnvA*, UniAP and Galvatron yield the same optimal strategy for BERT-Huge-32, T5-Large-48, and ViT-Huge-32, outperforming Alpa in terms of training throughput and strategy search time. In addition, when handling Swin-Huge-48, UniAP finds a solution while Galvatron encounters CUDA OOM issues. Notably, UniAP achieves a maximum search speedup that is 17× faster than Galvatron and hundreds of times faster than Alpa on BERT-Huge-32. This is primarily due to the ability of the MIQP solver to search for an optimal strategy with multiple threads, while the dynamic programming-based methods like Galvatron and Alpa run on a single thread because of their strong data dependency.

On *EnvB*, UniAP consistently demonstrates competitive or faster training throughput compared to Galvatron and Alpa. Simultaneously, UniAP's strategy searching time is also significantly shorter than these two baseline methods. Upon deeper examination of the parallel strategies discovered by Galvatron and Alpa, we find that UniAP stands out by identifying a superior solution with higher model FLOPs utilization (MFU) compared to Galvatron and Alpa. This is attributed to UniAP's broader strategy space, achieved by jointly optimizing inter- and intra-layer automatic parallelism

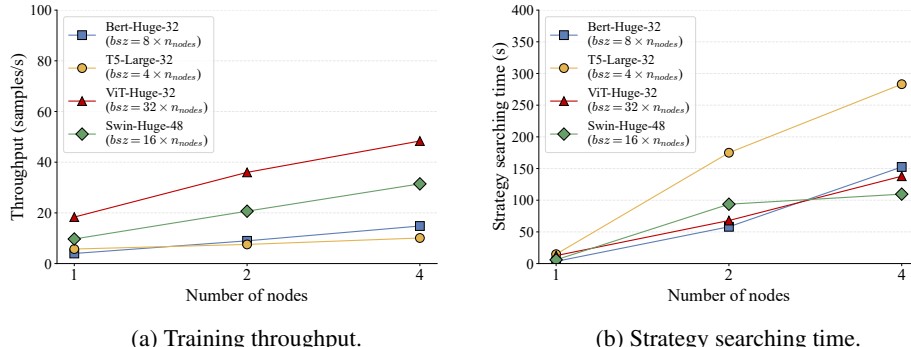

(a) Training throughput.                    (b) Strategy searching time.

Figure 4: Training throughput and strategy searching time with different number of nodes for different models. $n_{nodes}$ denotes the number of nodes for the corresponding data point and $bsz$ denotes the mini-batch size used for evaluation.

Table 2: Ablation study on the importance of unifying strategy space. SOL× indicates no solution during strategy searching, whereas CUDA× represents CUDA OOM exceptions during model training.

| Model | Training throughput (samples/s) | | |
|---|---|---|---|
| | UniAP (Inter-only) | UniAP (Intra-only) | UniAP |
| BERT-Huge-32 | SOL× | $2.48 \pm 0.02$ | $\mathbf{10.77 \pm 0.13}$ |
| T5-Large-32 | SOL× | $2.92 \pm 0.01$ | $\mathbf{9.01 \pm 0.06}$ |
| ViT-Huge-32 | $\mathbf{45.58 \pm 0.54}$ | CUDA× | $\mathbf{45.58 \pm 0.54}$ |
| Swin-Huge-48 | $\mathbf{19.08 \pm 0.10}$ | $4.66 \pm 0.02$ | $\mathbf{19.08 \pm 0.10}$ |

using a unified MIQP formulation. For further discussions, we provide a case study by visualizing the parallel strategy discovered by UniAP and MFUs for different approaches in Appendix E.

## 4.2 SCALABILITY

In this section, we conduct a scalability study on *EnvC* for UniAP. As Figure 4 shows, the training throughput of the optimal strategy and its strategy searching time on *EnvC* exhibits near-linearity in a real-world system as the number of nodes and mini-batch size increase exponentially. This phenomenon verifies the computational complexity analysis in Section 3.5.

## 4.3 ABLATION STUDY

In this section, we study the importance of strategy space on the optimality of parallel strategies. Specifically, we reduce the strategy space to inter-layer-only and intra-layer-only strategies in UniAP and evaluate the training throughput of the resulting optimal strategy on *EnvB*. We set the mini-batch size as 16, 12, 64, and 32, respectively. Table 2 shows that constraining the strategy space can compromise the optimality of parallel strategies or provide strategies that encounter OOM across different models. Therefore, holding a unified view of the automatic parallelism problem is essential.

## 5 CONCLUSION

In this paper, we propose UniAP, the first framework to conduct a joint strategy searching in inter- and intra-layer parallelism strategy space. Our experimental results show that UniAP speeds up model training on four Transformer-like models by up to $1.7\times$ and reduces the strategy searching time by up to $107\times$. Moreover, the optimal parallel strategies discovered by UniAP exhibit scalability on training throughput and strategy searching time.

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

## A    PROOF OF THE LINEAR FORM FOR THE CONTIGUOUS SET

To facilitate our discussion, we adopt the linear form of the contiguous constraint as presented in the main paper. We denote $P_{ui}$ as a 0-1 variable indicating whether layer $u$ is to be placed on the $i$-th computation stage, $deg$ as the number of computation stages in the pipeline. Besides, $\mathcal{G}(V, E)$ represents the computation graph for the model. Then, we formalize the theorem as follows:

**Theorem 1.** *A subgraph $V_i = \{\forall u \in V : P_{ui} = 1\}$ is contiguous if and only if there exists $Z_{vi}$ such that equation 11, equation 12, and equation 13 are satisfied.*

Previous work (Tarnawski et al., 2020) has proven this theorem. Our proof draws on the process of this work. The details of the proof are as follows:

**Proof 1.** *"If": Assume that there exists nodes $u, w \in V_i$ and $v \notin V_i$ such that $v$ and $w$ are reachable from $u$ and $v$, respectively. Hence, $P_{ui} = 1$, $P_{wi} = 1$, and $P_{vi} = 0$. Without losing generality, we assume $\langle u, v \rangle \in E$. Thus, according to equation 13, we have $Z_{vi} \leqslant P_{vi} - P_{ui} + 1 = 0$. By applying equation 12 repeatedly following the path from $v$ to $w$, we have $Z_{wi} \leqslant Z_{vi}$. Thus, $Z_{wi} \leqslant 0$. However, we also have $Z_{wi} \geqslant P_{wi} = 1$ according to equation 11. A contradiction.*

*"Only if": First, we define $Z_{vi} = 1$ if a node $w \in S$ is reachable from $v$. Otherwise, $Z_{vi} = 0$. Thus, equation 11 and equation 12 are satisfied according to this kind of definition. For equation 13, if $P_{vi} = 1$, the constraint will hold true regardless of whether $P_{ui}$ is 1 or 0. If $P_{vi} = 0$ and $P_{ui} = 0$, $Z_{vi} \leqslant P_{vi} - P_{ui} + 1 = 1$ will also hold true because $Z_{vi}$ could be either 0 or 1. Finally, if $P_{vi} = 0$ and $P_{ui} = 1$, $Z_{vi} = 0$ will hold true because $V_i$ is a contiguous set and we couldn't find any $w \in V_i$, such that $w$ is reachable from $v$.* □

## B    QIP FORMULATION FOR INTRA-LAYER-ONLY PARALLELISM

Here we present the QIP formulation for intra-layer-only parallelism with explanations.

$$\min \quad tpi = p_1, \tag{QIP}$$

$$\text{s.t.} \quad \sum_{u \in V} S_u^{\mathsf{T}} A_u + \sum_{\langle u,v \rangle \in E} S_u^{\mathsf{T}} R_{uv} S_v = p_1, \tag{14}$$

$$\sum_{u \in V} S_u^{\mathsf{T}} M_u \leqslant m, \tag{15}$$

$$\sum_{k=1}^{|g_u|} S_{uk} = 1, \qquad\qquad \forall u \in V, \tag{16}$$

$$S_{uk} \in \{0, 1\}, \qquad\qquad \forall u \in V, k \in \{1, \ldots, |g_u|\}. \tag{17}$$

The objective function equation QIP tends to minimize the TPI, thereby maximizing training throughput. This function solely takes the value of $p_1$ into account, as there is only one computation stage involved in the intra-layer-only parallelism. Subsequently, we proceed to explain the constraints of this formulation:

- Equation 14 encodes the intra-layer-only computation and communication costs as $p_1$. The first summation term for any $u \in V$ represents the cost of choosing intra-layer strategies for all layers. The second term represents the summation of resharding costs on all edges.

- Equation 15 encodes that the memory consumption on a single device should not exceed its device memory bound $m$. It is worth noting that $m$ should be an identical constant across multiple devices if these devices are homogeneous. Otherwise, the value of $m$ varies.

- Equation 16 and equation 17 indicate that each layer should select exactly one strategy.

## C    COMPLEXITY ANALYSIS

Let $|V|$, $|g|$, and $n$ denote the number of layers, parallel strategies, and the number of GPUs, respectively. As illustrated in Algorithm 1, UniAP searches all possible pipeline stages exponentially

Table 3: Details for four Transformer-like models.

| Model | Layers | Hidden size | Sequence length | Parameter size |
|-------|--------|-------------|-----------------|----------------|
| BERT-Huge | 32 | 1280 | 512 | 672M |
| T5-Large | 16/16 | 1024 | 512 | 502M |
| ViT-Huge | 32 | 1280 | 196 | 632M |
| Swin-Huge | 2/2/42/2 | 320/640/1280/2560 | 49*64/49*16/49*4/49*1 | 1.02B |

until $n$ is reached. Given a hyperparameter of mini-batch size $B$, UniAP invokes `CalculateCost` to model each layer's costs for each parallel strategy. Additionally, the optimization time limit of the MIQP solver can be set as a constant hyperparameter when UniAP calls it. Therefore, the overall computational complexity of UniAP is $\mathcal{O}(|V||g|\log(n))$.

## D  EXPERIMENT DETAIL

**Gurobi configuration**   When tackling the MIQP problem, UniAP employs several configurations for the Gurobi Optimizer 10.1 (Gurobi Optimization, LLC, 2023). In particular, we set *TimeLimit* to 60 seconds, *MIPFocus* to 1, *NumericFocus* to 1, and remain other configurations to default. For instance, we establish the *MIPGap* parameter as the default value of 1e-4 to serve as a strict termination criterion. Furthermore, we have implemented an early stopping mechanism to terminate the optimization process as early as possible. There are two conditions that can activate the mechanism. Firstly, if the current runtime exceeds 15 seconds and the relative MIP optimality gap is less than 4%, we will terminate the optimization. Secondly, if the current runtime exceeds 5 seconds and the best objective bound is worse than the optimal solution obtained in the previous optimization process, we will terminate the optimization.

**Model detail**   Table 3 presents the details of four Transformer-like models selected for our evaluations. Two of these models, namely BERT-Huge (Devlin et al., 2019) and T5-Large (Raffel et al., 2020), belong to the domain of natural language processing (NLP). At the same time, the remaining two, ViT-Huge (Dosovitskiy et al., 2021) and Swin-Huge (Liu et al., 2021), are associated with computer vision (CV). It is noteworthy that BERT-Huge and ViT-Huge share the same hidden layer type, whereas T5-Large and Swin-Huge have multiple layer types. Numbers separated by slashes represent the statistical information for different layer types. For instance, Swin-Huge comprises four types of layers, each with 2, 2, 42, and 2 layers, respectively.

**Training detail**   UniAP is based on the PyTorch framework and integrates models from Hugging-Face Transformers. It employs various types of parallelism, including Pipeline Parallelism (PP), Data Parallelism (DP), Tensor Parallelism (TP), and Fully Sharded Data Parallelism (FSDP), utilizing GPipe (Huang et al., 2019), PyTorch DDP (Li et al., 2020), Megatron-LM (Narayanan et al., 2021b), and FairScale (FairScale authors, 2021), respectively. For NLP models, we use the English Wikipedia dataset (Wikimedia Foundation, 2023), while the ImageNet-1K dataset (Russakovsky et al., 2015) is used for CV models. We train these models using the Adam optimizer (Kingma & Ba, 2015) and precision of FP32. We omit hyperparameters here such as learning rate and weight decay as these have minimal impact on training throughput. The model parameters in the HuggingFace Transformer are configured to align with the specifications of each individual model. For instance, we set *hidden_size* to 1280, *num_hidden_layers* to 32, *num_attention_heads* to 16, and *seq_length* to 512 for BERT-Huge. Regarding other hyperparameters in the HuggingFace configurations, we set *hidden_dropout_prob* and *attention_probs_dropout_prob* to 0.0 for ViT-Huge. For Swin-Huge, we set *drop_path_rate* to 0.2. We remain other configurations to default. It should be noted that the training batch sizes for each experiment are outlined in the main paper.

## E  CASE STUDY: BERT-HUGE

In this section, we present a visualization of the optimal parallelism strategy discovered by UniAP. As represented in Figure 5, the strategy pertains to training BERT-Huge with 32 hidden layers in

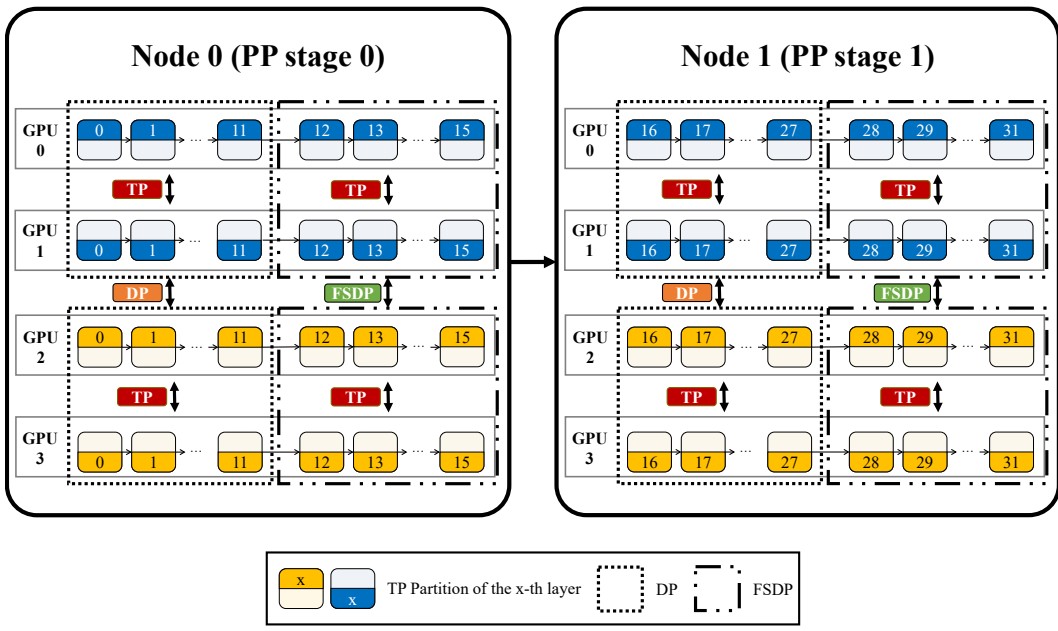

Figure 5: The optimal parallel strategy for all hidden layers of BERT-Huge on *EnvB*. Different colors represent different input samples in a micro-batch.

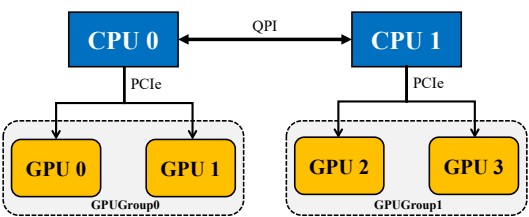

Figure 6: Topology of a node in *EnvB*.

a 2-node environment *EnvB* with a mini-batch size of 16. Each node was equipped with 2 Xeon E5-2620 v4 CPUs, 4 TITAN Xp 12GB GPUs, and 125GB memory. These nodes are interconnected via a 10Gbps network. It should be noted that we only showcase the parallelism strategy for the hidden layers here for simplicity but without losing generality.

Here, we provide further topological information for a node within *EnvB*. As illustrated in Figure 6, we categorize the GPUs numbered 0 and 1 in each node and refer to them collectively as *GPUGroup0*. Similarly, we label the GPUs numbered 2 and 3 as *GPUGroup1*. In *EnvB*, the interconnects within each GPU group (i.e., PCIe) have superior bandwidth than that between different groups (i.e., QPI). We collectively designate these two connection bandwidths as intra-node bandwidth, which is higher than inter-node bandwidth.

In this example, UniAP has identified a parallelism strategy for inter-layer parallelism that involves a two-stage pipeline. This strategy utilizes parallelism in a manner that is both efficient and effective. Specifically, the communication cost of point-to-point (P2P) between two nodes is less than that of all-reduce. Additionally, the inter-node bandwidth is lower than that of the intra-node. These factors make the two-stage PP approach a reasonable choice. Moreover, the pipeline has been designed such that each stage comprises an equal number of layers. This design leverages the homogeneity of the nodes and ensures load balancing across the cluster.

UniAP employs an intra-layer parallelism strategy within each PP stage. It utilizes a 2-way DP for the initial 12 hidden layers in each stage between *GPUGroup0* and *GPUGroup1*. For the remaining four hidden layers, a 2-way FSDP is utilized between *GPUGroup0* and *GPUGroup1* to reduce memory footprint and meet memory constraints. Within each GPU group, UniAP employs a 2-way TP for each

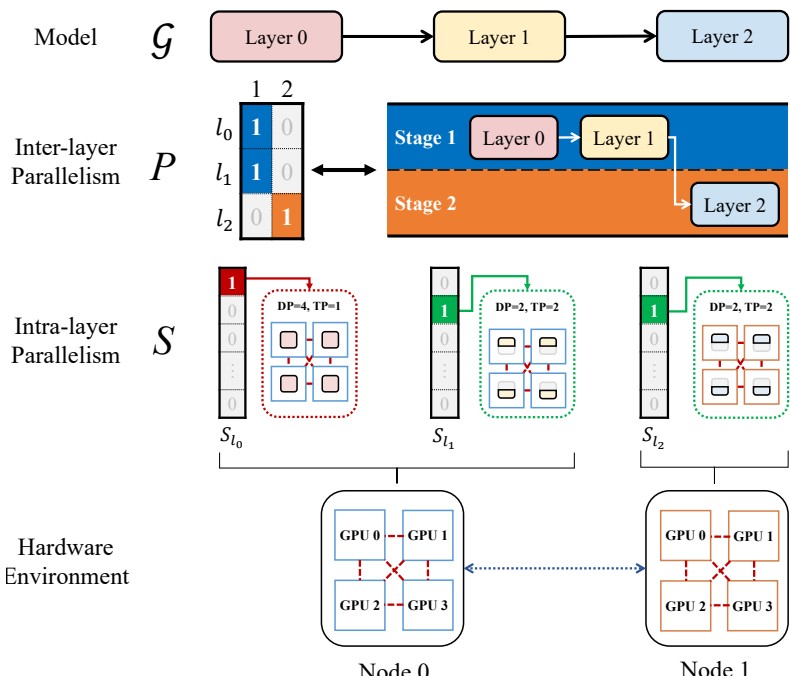

Figure 7: A candidate solution for MIQP.

layer. In general, TP incurs more significant communication volumes than DP and FSDP. In order to achieve maximum training throughput on *EnvB*, it is necessary to implement parallelism strategies that prioritize higher communication volumes within each group and lower volumes between groups. Therefore, the strategy for BERT-Huge with 32 hidden layers combines the best elements of PP, DP, TP, and FSDP to maximize training throughput.

In addition, we have conducted calculations for the model FLOPs utilizatio (MFU) for Galvatron, Alpa, and UniAP in this scenario to validate our analysis. MFU is a metric introduced by Chowdhery et al. (2023), which is independent of hardware, frameworks, or implementations. Therefore, it allows us to examine the performance of different parallel strategies solely from a strategic perspective. For BERT-Huge-32, the resulting MFUs for UniAP, Galvatron, and Alpa on *EnvB* are 23.6%, 13.7% and 19.6% , respectively. Therefore, we conclude that UniAP does utilize its larger strategy space to find a globally optimal solution, rather than a locally optimal one.

## F   VISUALIZATION FOR THE CANDIDATE SOLUTION

In this section, we proceed to visually represent a potential solution for equation MIQP. Given a deep learning model $\mathcal{G}$, UniAP will determine the placement strategy $P$ for inter-layer parallelism and the parallel strategy $S$ for intra-layer parallelism using an off-the-shelf solver. As depicted in Figure 7, the solver is optimizing a 3-layer model with two pipeline stages, each assigned 4 GPUs. Throughout this process, a potential solution could be

$$P = \begin{bmatrix} 1 & 0 \\ 1 & 0 \\ 0 & 1 \end{bmatrix}, \ S = \begin{bmatrix} 1 & 0 & 0 \\ 0 & 1 & 1 \\ 0 & 0 & 0 \\ \vdots & \vdots & \vdots \\ 0 & 0 & 0 \end{bmatrix}. \tag{18}$$

Here, the $u$-th row of matrix $P$ denotes the placement strategy for layer $u$, where $P_{ui} = 1$ signifies the placement of layer $u$ on stage $i$, while $0$ indicates otherwise. For example, $P_{l_0} = [1, \ 0]$ denotes the placement of layer $l_0$ on pipeline stage 1. Additionally, the $u$-th column of matrix $S$ the selected

Table 4: Relative estimation error on the performance modeling.

| Env. | Model | Training throughput (samples/s) | | $e_{relative}$ (%) |
|------|-------|-----------|--------|----------------|
| | | Estimated | Actual | |
| *EnvA* | BERT-Huge-32 | 34.96 | $33.46 \pm 0.28$ | $4.46 \pm 0.83$ |
| | T5-Large-48 | 21.33 | $23.296 \pm 0.04$ | $8.49 \pm 0.17$ |
| | ViT-Huge-32 | 109.76 | $109.51 \pm 0.068$ | $0.23 \pm 0.06$ |
| | Swin-Huge-32 | 67.54 | $68.80 \pm 0.12$ | $1.83 \pm 0.18$ |
| *EnvB* | BERT-Huge-32 | 10.17 | $10.77 \pm 0.13$ | $5.59 \pm 1.17$ |
| | T5-Large-32 | 8.24 | $7.98 \pm 0.05$ | $3.29 \pm 0.56$ |
| | ViT-Huge-32 | 43.91 | $45.58 \pm 0.54$ | $3.67 \pm 1.19$ |
| | Swin-Huge-32 | 19.29 | $19.08 \pm 0.10$ | $1.12 \pm 0.54$ |

intra-layer parallelism strategy for layer $u$, where $S_{uj} = 1$ denotes the selection of the $j$-th strategy from the intra-layer parallelism strategy set. For example, $S_{l_0} = [1, 0, 0, \cdots, 0]^\top$ indicates that layer $l_0$ will adopt the pure DP strategy, while $S_{l_1} = [0, 1, 0, \cdots, 0]^\top$ indicates that layer $l_1$ will employ a strategy where DP is performed on GPU 0, 1 and GPU 2, 3, and TP is performed across these two GPU groups.

There exist numerous combinations of $P$ and $S$. The off-the-shelf solver will automatically search for the optimal configuration. By integrating these inter- and intra-layer strategies, along with the pipeline stages and chunk sizes enumerated in the outer UOP process, UniAP will ultimately derive a globally optimal parallel strategy for the deep learning model within the current hardware environment.

# G    EVALUATION OF PERFORMANCE MODELING

Accurate performance modeling for time-per-iteration (TPI) is crucial for evaluating candidate strategies and identifying a globally optimal one. To this end, we compare the simulated throughput of the optimal parallel strategy with the actual throughput obtained by testing the strategy on two environments, namely *EnvA* and *EnvB*. In order to quantify the accuracy of the estimated training throughput, we introduce a metric called relative estimation error $e_{relative}$, which is computed using the following equation:

$$e_{relative} = \frac{|throughput_{actual} - throughput_{estimated}|}{throughput_{actual}} \times 100\% \tag{19}$$

Here, $throughput_{estimated}$ denotes the training throughput estimated by the automatic parallelism, while $throughput_{actual}$ denotes the actual throughput profiled during the training process.

Our results, as shown in Table 4, demonstrate that UniAP achieves an average relative estimation error of approximately 3.59%. In contrast, the average relative estimation error for Galvatron in our experiments is 11.17%. Given that UniAP requires significantly less time for strategy searching compared to other approaches, as depicted in Table 1, we can conclude that UniAP's performance modeling strikes a good balance between accuracy and efficiency.

