# OpenReview forum: "UniAP: Unifying Inter- and Intra-Layer Automatic Parallelism by Mixed Integer Quadratic Programming"
_ICLR.cc/2024/Conference — Submitted to ICLR 2024_

### Official Review · Reviewer_Ye4S · 2023-11-01

**Soundness:** 3 good
**Presentation:** 3 good
**Contribution:** 3 good
**Rating:** 6
**Confidence:** 3

**Summary:**

This paper presents a mixed integer quadratic programming (MIQP) based approach for automatic parallelization in transformer training, jointly optimizing both inter-layer and intra-layer parallelism. UniAP formulates the total time-per-iteration (TPI) as a quadratic function of the decision variables that indicate the intra-layer parallelization strategies (DP-Data parallelism, TP-tensor parallelism, FSDP-fully sharded data parallelism) and the pipeline stage partitioning strategies. The experiments show that UniAP achieves competitive or better performance with Alpa and Gavaltron on transformer-based models.

**Strengths:**

1. This work comprehensively optimizes intra-layer (DP, TP, and FSDP) and inter-layer(PP) partitioning decisions under one unified view. An advantage of this approach is that it can jointly optimize both targets. By doing so, it avoids the potential issue of optimizing one factor at the expense of the others like the bi-level optimization approach.
3. The evaluation highlights the effectiveness of the formulation, with improved training performance and faster search time achieved using an off-the-shelf MIQP solver compared to a previous dynamic programming and greedy algorithm-based approach Galvatron or Alpa.

**Weaknesses:**

1. Since Alpa already uses ILP to formulate the intra-operator parallelism, it may be straightforward to extend it to MIQP since MIQP is more like a variant of ILP.
2. The work makes certain assumptions about performance that are not always true. For instance, in Section 3.3, it states that "For Transformer-like models that mainly consist of the MatMul operator, the computation time in the BP stages is roughly twice that of the FP stages", which may not always be true depending on the optimizer
3. Regarding the comparison to Alpa, the evaluation is not apple-to-apple. Alpa builds on JAX which uses XLA for JIT optimization, while Gavlaltron uses `torch.compile` for JIT. It would be best to compare Alpa and UniAP on the same framework, possibly done by porting to JAX.

**Questions:**

1. The pipeline parallelism formulation of UniAP was based on a GPipe-style schedule, which has a major drawback of large bubble (idle) time. Would it be possible to adapt to more advanced PP schedules like DAPPLE or Chimera, and how would that affect your formulation?
2. It states that UniAP "partitions stages and determines micro-batch size using naive greedy algorithm". Could you elaborate on the exact algorithm you used, and whether you have reported the time for searching the microbatch+#PP stage in the search time (Table 1&2)?

---

> ### Author Response · Authors · 2023-11-20
>
> We sincerely thank you for your detailed review. Please find our responses below.
>
> Weaknesses:
>
> W1: It is straightforward to extend ILP to MIQP, which looks like a variant of ILP.
>
> R1: From a formulation perspective, ILP is a special case of MIQP. However, for automatic parallelism problems, MIQP cannot be considered as a simple generalization of ILP due to several reasons.
>
> One of the most significant reasons is that ILP can only address intra-layer parallelism, while it cannot handle the quadratic term introduced by the resharding cost in the computation graph, which is necessary for formulating unified inter- and intra-layer parallelism problems.
>
> Additionally, there are various styles of inter-layer parallelism (or pipeline parallelism), and it is challenging to formulate the inter-layer cost for different pipeline schedules while maintaining consistency in the formulation. This difficulty requires a higher-level examination of different pipelines, including different memory consumptions and bubble ratios.
>
> Finally, unlike intra-layer parallelism, inter-layer parallelism is not based on the SPMD programming model. (Xu et. al., 2021, and additional reference Wang et. al., 2023). Thus, we need to accommodate the MIQP formulation into the non-SPMD programming model, which leads to the Order-Preserving constraint (Equation 5) to prevent disordered layer assignments.
>
> Overall, due to our thorough abstraction and modeling, the MIQP expression is simple and uniformly expresses the complex problem of inter- and intra-parallel strategy search. Therefore, MIQP is not a straightforward extension from ILP for the automatic parallelism problem.
>
>
> W2: Assumption "the computation time in the BP stages is roughly twice that of the FP stages" may not always be true because the optimizer varies.
>
> R2: The assumption "the computation time in the BP stages is roughly twice that of the FP stages" is for estimating the overhead for backward computation time, which has been commonly adopted in distributed learning (Narayanan et al., 2021b; Li & Hoefler, 2021; Miao et al., 2022). UniAP follows this approach. It is important to note that the assumption is a trade-off for accuracy and fast cost modeling. Techniques such as modeling backward computation time for specific optimizers or different operators are not the current focus and are left for our future work.
>
> W3: Evaluation is not apple-to-apple.
>
> R3: We acknowledge your opinion but assure you that we have strived to ensure fairness between Alpa and UniAP. Porting UniAP to JAX required rebuilding the entire profiling process, TP, PP, and FSDP module wrapper, communication primitives for specific parallelism schemes, and extensive work in debugging, verifications, and alignments. We were unable to meet our expectations, and we hope for your understanding.
>
> However, we have revised the MFU results for our case study in the last paragraph of Appendix E. The metric is hardware and implementation independent. The results indicate that UniAP can achieve an MFU of approximately 23.6%, while Galvatron and Alpa can only achieve 13.7% and 19.6%, respectively. This may help to explain that UniAP's superior performance is solely due to its globally optimal parallel strategy.
>
> Questions:
>
> Q1: Would it be possible for more advanced PP schedules? How would that affect the formulation?
>
> A1: More advanced PP schedules are possible. For example, the synchronous 1F1B pipeline proposed in PipeDream-2BW and DAPPLE maintains the MIQP's objective function due to the same bubble ratio as GPipe but offers reduced memory usage. To accommodate this change, one needs to adjust the constant coefficient for the activation memory term in the memory cost model. This illustrates the MIQP algorithm's flexibility in adapting to various PP strategies with an if-else clause.
>
>
> Q2.1: What is the exact algorithm for searching the micro-batch size and #PP stage?
>
> A2.1 The method "partitions stages and determines micro-batch size using a naive greedy algorithm" is adopted by Galvatron rather than UniAP. UniAP exhaustively searches for micro-batch size + #PP stages, as described in section 3.5.
>
> Q2.2: Whether you have reported the time for searching the microbatch + #PP stage in the search time?
>
> A2.2: Yes, we have reported the time for searching the micro-batch + #PP stage in the overall search time by "measuring the time of the UOP" as described in the experimental section in section 4.
>
> Additional references
>
> 1. 	Siyu Wang, Zongyan Cao, Chang Si, Lansong Diao, Jiamang Wang, Wei Lin. Ada-Grouper: Accelerating Pipeline Parallelism in Preempted Network by Adaptive Group-Scheduling for Micro-Batches. CoRR abs/2303.01675 (2023).

---

> > ### Comment · Reviewer_Ye4S · 2023-12-05
> >
> > Thank you for the detailed and thorough response!

---

### Official Review · Reviewer_S34h · 2023-11-01

**Soundness:** 3 good
**Presentation:** 4 excellent
**Contribution:** 3 good
**Rating:** 6
**Confidence:** 4

**Summary:**

This paper presents an automated parallelism framework that comprehensively considers both inter- and intra-parallelism.  In comparison to other parallel frameworks, this framework offers a larger strategy space. Additionally, a method based on Mixed Integer Quadratic Programming (MIQP) is introduced to obtain the optimal strategy within this strategy space. Through experiments, the proposed approach demonstrates a 1.7x throughput acceleration compared to other STOA parallel frameworks and a 261x improvement in strategy search efficiency.

**Strengths:**

1. This article raises a meaningful question and offers a reasonable solution.
2. This article is well-written

**Weaknesses:**

1. How is the accuracy of your cost model?
2. Section 3.4 formalizes the MIQP; however, it does not incorporate constraints related to computing resources. More precisely, there is no restriction on the number of GPUs employed by the strategy, thereby allowing it to stay within the available GPU count.
3. What is the form of the inter-parallel strategy? Is it a simple selection from DP  TP FSDP, or is it a more complex form?
4.  The maximum parameter size of the models used in the experimental section is 1.02 billion. For the current trend in the development of large models, these models are slightly smaller. We look forward to the authors presenting the parallel performance of larger models. Perhaps your available computing resources limit the use of larger models, and in that case, simulated results would also be acceptable.

**Questions:**

1. How is the accuracy of your cost model?
2. Section 3.4 formalizes the MIQP; however, it does not incorporate constraints related to computing resources. More precisely, there is no restriction on the number of GPUs employed by the strategy, thereby allowing it to stay within the available GPU count.
3. What is the form of the inter-parallel strategy? Is it a simple selection from DP TP  FSDP, or is it a more complex form?
4.  The maximum parameter size of the models used in the experimental section is 1.02 billion. For the current trend in the development of large models, these models are slightly smaller. We look forward to the authors presenting the parallel performance of larger models. Perhaps your available computing resources limit the use of larger models, and in that case, simulated results would also be acceptable.

---

> ### Author Response · Authors · 2023-11-20
>
> Thank you for your thorough review. Please find our responses below.
>
> Weaknesses:
>
> W1: The accuracy of the cost model.
>
> R1: We have included end-to-end (E2E) assessments of estimated throughput accuracy in Appendix G. In our experiments, the average relative estimation error is 3.59%, significantly lower than Galvatron's 11.17%. This outcome showcases the robustness of UniAP's performance modeling.
>
> W2: Lack of restrictions on the number of GPUs.
>
> R2: The number of GPUs has been implicitly included in the strategy matrix $S$. We enumerate all possible combinations of intra-layer strategies for each layer given the partitioned pipeline stages. Each pipeline stage has a designated number of GPUs, and thus the computing resources are well constrained.
>
>
> W3: The form of inter-layer parallel strategy.
>
> R3: The form of the inter-layer parallel strategy is pipeline parallelism (PP), consistent with previous work (Zheng et al., 2022).
>
> W4: The maximum parameter size of the models is small.
>
> R4: Due to limited computing resources, we are unable to test larger models. However, our evaluation of end-to-end (E2E) throughput estimation against actual results shows that the E2E estimation error is limited to 3.59% in our environments. This observation highlights the potential of UniAP in large-scale clusters. Please refer to C2 and R2 in General Responses for further elaborations.
>
>
> Questions:
>
> All questions have been addressed in R1, R2, R3, and R4, respectively. Please find our responses above.

---

### Official Review · Reviewer_pnV8 · 2023-11-03

**Soundness:** 2 fair
**Presentation:** 2 fair
**Contribution:** 2 fair
**Rating:** 5
**Confidence:** 4

**Summary:**

The paper presents UniAP, a system for automatically parallelizing deep learning training workloads across devices. Compared to prior work in the area (e.g., Alpa), the key point is unifying inter and intra-operator parallelism within a single formulation for determining a parallel strategy. Experimental evaluation shows using UniAP delivers improved performance.

**Strengths:**

1. UniAP aims to improve automatic parallelization for training, which has wide practical benefits for researchers and practitioners. Although not mentioned in the paper, I would add that it seems like it could be particularly beneficial for fine-tuning, which is often performed in situations with fewer resources than full training from scratch.
2. Further, UniAP aims to unify the representation of parallelization strategies into a single formulation for optimization, a nice result.
3. The experiments include a variety of transformer and vision transformer models and demonstrate improved performance over existing methods (e.g., Alpa).

**Weaknesses:**

1. I found the paper somewhat hard to follow, and details are unclear. This is especially the case for the formulation of the actual optimization problem, which would benefit from an explanatory figure (and possibly a table of notation). There are also important details that are elided, such as the communication model (the paper states "UniAP estimates the communication time by dividing the size of transmitting tensors by the communication efficiency for different communication primitives"); what exactly is used as the "communication efficiency"? This seems especially unclear as common libraries (e.g., NCCL) implement different algorithms for the same operation, which can have very different performance (e.g., logarithmic versus linear terms in tree versus ring allreduces).
2. The paper claims to unify inter- and intra-layer parallelism, but it does not seem that way to me. In particular, UniAP appears to define a cost model which can be used to evaluate the performance of a configuration, but then exhaustively enumerates different pipeline configurations (Section 3.5 & Algorithm 1) in order to find the one with least cost. The paper should address this, and also discuss how this approach compares to those of prior works (e.g., the dynamic programming formulation in Alpa).
3. The experiments are conducted at quite small scale; the largest cluster (EnvC) has only four nodes and 16 GPUs. Meanwhile, the models are also quite small, with the largest being 1 B parameters (Table 4). While this might be plausible for a fine-tuning situation (although even there one should expect larger models), this seems unrealistic for training large models from scratch, a scenario the paper seems to be targeting. I would like to see experiments with both larger models and clusters (as a data point, the Alpa paper includes results on models up to 70 B parameters). Indeed, it may be best to add comparisons using the same models as in the Alpa paper to provide a solid baseline.
4. A related point, given the search over pipeline configurations, it is not clear to me how UniAP's search would scale. For example, how would it perform if the cluster contained 10 000 GPUs? (This is not at all unrealistic for large training clusters.)
5. The experiments omit error bars or any measure of variance, making it hard to judge how stable the results are.
6. The paper omits important techniques used in practice from its evaluation, in particular, mixed precision (Section 4). While I appreciate the claim that this is orthogonal, using FP32 can result in significantly lower computation performance (due to not using tensor cores), which in turn can significantly change the communication/computation ratio and consequently the best parallel strategy. This is especially strange because Alpa evaluated transformers in FP16.
7. The experimental evaluation lacks depth. There is not detailed discussion of _why_ UniAP outperforms prior solutions. For example, one could look at the resulting communication volume or GPU utilization. It is hard to disambiguate whether the improved performance is due to framework/configuration differences, or due to improved parallel strategies found by UniAP. Detailed profiling and analysis is necessary. While the paper does briefly mention some differences in the parallel strategies found, these are not discussed in detail.

**Questions:**

Please see the above for details.

1. Please improve the clarity of the paper, particularly regarding the presentation of the search algorithm.
2. Please clarify how UniAP "unifies" intra- and inter-layer parallelism search and how its method is superior to those in prior works?
3. Please conduct experiments at larger scale and with larger models. How does UniAP's search scale with larger clusters?
4. Please add error bars to all experimental results.
5. Why do you not use AMP in the evaluation? If you do, what changes?
6. Please provide a detailed analysis of _why_ UniAP's parallel strategies outperform those of other frameworks.

-----

In light of the authors' response, I have raised my score; however, I still have concerns about the lack of larger-scale evaluation, as such environments are where automatic parallelism is most promising.

---

> ### Author Response · Authors · 2023-11-20
>
> Thank you for your time and valuable feedback. Please find our responses below.
>
> Weaknesses:
>
> W1: Hard to follow and details are unclear.
>
> W1.1: The formulation for the actual optimization problem lacks explanatory figure.
>
> R1.1: We have included an additional figure in the appendix to help explain the optimization problem. Please refer to it, along with the supplemental descriptions in Appendix F.
>
> W1.2: Important details for communication model are elided.
>
> R1.2: We appreciate your feedback regarding the term "communication efficiency." We will revise it to "profiled communication efficiency" for better clarity. In the context of DP, TP, and FSDP, there are three main CCs: all-reduce, all-gather, and reduce-scatter. Each of these algorithms has its own complexity under different circumstances, but they are all optimized to suit specific hardware or networks. Our paper currently focuses on the parallel strategy search process and treats the specific collective algorithm chosen by the user or backend as a black box. We view this as a trade-off between accuracy and complexity. We have conducted an end-to-end (E2E) evaluation on our performance modeling in Appendix G. In the current evaluation, the average relative estimation error for UniAP is approximately 3.59%, which is lower than that of Galvatron.
>
> W2: See W2.1 and W2.2.
>
> W2.1: Misclaim on unifying inter- and intra-layer parallelism.
>
> R2.1: Unifying inter- and intra-layer parallelism and finding one strategy with minimum cost are two sides of the same coin. To jointly optimize for a globally optimal inter- and intra-layer parallelism strategy, we set our goal to selecting an inter- and intra-layer strategy with maximum throughput, or in other words, minimum time-per-iteration (TPI). Thus, we have to model the time cost for each possible inter- and intra-layer parallelism strategy, as well as the memory cost to avoid out-of-memory (OOM) exceptions. Therefore, the cost model, the enumeration process in UOP, and so on, is our proposed solution to jointly optimize for a globally optimal strategy.
>
> W2.2: How does UniAP compare to prior works?
>
> R2.2: Existing methods do not optimize these strategies jointly from a global perspective. For instance, Alpa feeds the ILP results for intra-operator stages to inter-operator stages for an optimal solution. However, combining local optima doesn't always ensure a global optimum. In certain cases, two consecutive operators with different optimal intra-operator sharding specs may result in higher communication costs between them. In systems with PCIe interconnections, it may be globally optimal to compromise on one of them to keep them the same, as it avoids invoking any CCs. Unfortunately, Alpa is unable to identify such cases as it only returns a single solution rather than a series of candidates after each intra-operator pass. This is where UniAP provides a unique solution.
>
> Apart from Alpa, existing methods either overlook potential opportunities for inter- or intra-layer parallelism or optimize hierarchically instead of globally. They do not unify the complete inter- and intra-layer parallel strategy space to jointly search for a globally optimal solution. Based on these motivations, we identify two key challenges: formulating the problem globally and optimizing the formulation efficiently. To address these challenges, we extensively evaluated various existing methods and ultimately developed MIQP. To the best of our knowledge, none of the methods considered such an extensive strategy space and resolved the aforementioned challenges at the same time. Hence, UniAP distinguishes itself from prior works.
>
> W3: Experiments are conducted at quite small scale.
>
> R3: Due to the resource constraint in our lab which is from university, we regret not having a large cluster to conduct experiments on larger models (e.g., models with 70B parameters in Alpa). Simulated results are also infeasible due to the lack of profiled data on a large cluster to consolidate our MIQP optimization. However, to the best of our knowledge, the scale of clusters (*EnvA*, *EnvB* and *EnvC*) in our experiment is almost the largest one for most universities. Meanwhile, many vision models have about 1B parameters. Therefore, optimizing parallel strategies for efficiently training a 1B model is also meaningful and our method provides a potential way for doing research on large language models for these universities. Furthermore, the scale of the cluster is not a strong assumption for UniAP. On the contrary, UniAP scales well based on our theoretical analysis in Appendix C and preliminary experimental results in Section 4.2. Hence, we acknowledge the limitation of not including results on larger scale clusters, but for a university lab, we have made our best effort to achieve a relatively promising result.

---

> ### Author Response · Authors · 2023-11-20
>
> W4: How would UniAP's search scale over pipeline configurations and 10,000 GPUs, for example?
>
> R4: The search process will scale in the time complexity of $\mathcal{O}(|V||g|\log(n))$ as analyzed in Appendix C. Take $n=10,000$ GPUs as an example, the number of hidden layers $|V|$ remains a constant as it is orthogonal to the cluster size. Since 10,000 is not an exponent of 2, we enumerate its 25 factors to achieve load balance, resulting in a total of 2690 intra-layer parallelism candidates. Hence, the time complexity of $n=10,000$ GPUs would be $\mathcal{O}(2690|V|)$.
>
> W5: Omit error bars or any measure of variance.
>
> R5: We have included error bars and measures of variance in our revision. Please refer to Table 1 on pp.8, Table 2 on pp.9, and Table 4 on pp. 18.
>
> W6: Omit mixed precision in experiments.
>
> R6: In our paper, our focus is on developing a method to find the best hybrid parallel strategy, rather than seeking the best throughput across all existing training strategies. Therefore, enabling FP16, BF16, INT8, as well as activation checkpointing, gradient checkpointing, and other training techniques are irrelevant to our claims and contributions. However, these techniques are quite common in today's big model training, as you mentioned, and we will leave them for our future work. Regarding the experiments conducted on Alpa, we chose to use FP32 for comparison because FP16 was disabled in Galvatron's original paper. To maintain consistency and provide convincing results across the two baseline systems, we chose to align the floating point precision to 32 bits.
>
> W7: Experimental evaluation lacks depth.
>
> R7: Due to the limitation of 9 pages, we were unable to discuss why UniAP outperforms prior solutions in the main section. We have attempted to address the reasonableness of the solution UniAP finds in Appendix E. Additionally, to exclude potential impact on frameworks or configurations, we introduce the model FLOPs utilization (MFU) as an evaluation metric in our revision. MFU is hardware and implementation independent and depicts the GPU utilization ratio during model training. Our results for BERT-Huge on *EnvB* show that UniAP achieves an MFU of approximately 23.6%, higher than 13.7% for Galvatron and 19.6% for Alpa. Along with the intuitive analysis of the specific case, we conclude that UniAP benefits from its larger strategy space to come up with a globally optimal parallel strategy. Please refer to Appendix E for more details.
>
> Questions:
>
> Q1: Please improve the clarity of the paper, particularly regarding the presentation of the search algorithm.
>
> A1: We have updated "communication efficiency" to "profiled communication efficiency" and added a visualization of a candidate solution in Appendix F to clarify the optimization problem.
>
> Q2: Please clarify how UniAP "unifies" intra- and inter-layer parallelism search and how its method is superior to those in prior works?
>
> A2: Please refer to R2.1 and R2.2 for details.
>
> Q3.1: Please conduct experiments at larger scale and with larger models.
>
> A3.1: Please refer to R3 for details.
>
> Q3.2: How does UniAP's search scale with larger clusters?
>
> Q3.2: Details on how UniAP's search scales with larger clusters are provided in R4.
>
> Q4: Please add error bars to all experimental results.
>
> A4: Error bars have been added to our revision. Please check Table 1 on pp.8, Table 2 on pp.9, and Table 4 on pp. 18.
>
> Q5.1 Why do you not use AMP in the evaluation?
>
> A5.1 Please refer to W6 and R6 above.
>
> Q5.2 If you use AMP, what will change?
>
> A5.2 If we opt to enable AMP, we need to re-profile the computation time in AMP and rerun the strategy searching process (i.e. UOP). Apart from these steps, no more changes are needed.
>
> Q6: Please provide a detailed analysis of why UniAP's parallel strategies outperform those of other frameworks.
>
> A6: This has been primarily addressed in R7 above. Furthermore, we have updated our evaluation of training throughput and strategy searching time in Section 4.1. For a more comprehensive discussion, please refer to Section 4.1 and Appendix E.

---

> > ### Comment · Reviewer_pnV8 · 2023-11-22
> > **Response**
> >
> > Thank you for the extensive response. I have raised my score, but I still think larger-scale experiments (in both model and cluster size) are needed to really show the benefit of the method. I am sympathetic to issues of access to large-scale compute; could offloading (e.g., ZeRO-Infinity) be used to at least evaluate larger models (as in a fine-tuning setting)?
> >
> > > resulting in a total of 2690 intra-layer parallelism candidates
> >
> > Roughly how long would that take, in practice? I'm not fully confident in extrapolating from the values in Table 1. That said, for long training runs, spending a few hours on strategy search is not a big deal.
> >
> > > Our results for BERT-Huge on EnvB show that UniAP achieves an MFU of approximately 23.6%, higher than 13.7% for Galvatron and 19.6% for Alpa.
> >
> > Thanks for this, this is an improvement. However, I am still concerned, since all of these seem to be quite low compared to well-optimized implementations. For example:
> > * Ivanov et al., "Data Movement Is All You Need: A Case Study on Optimizing Transformers", MLSys 2021, achieves 35% of peak on V100s.
> > * Narayanan et al., "Efficient Large-Scale Language Model Training on GPU Clusters Using Megatron-LM", Supercomputing 2021, achieves 52% of peak on A100s.
> >
> > Are these baselines well-optimized?

---

> ### Author Response · Authors · 2023-11-23
>
> Thank you for your valuable feedback and understanding of our computing resource limitations. Please find our responses below.
>
> Q1: Could offloading (e.g., ZeRO-Infinity) be used to evaluate larger models?
>
> A1: Techniques such as offloading are independent of parallel training strategies. Enabling offloading will decrease the training throughput, which in turn will impact the fairness of comparing parallel strategies between different methods. Therefore, offloading may not be suitable for evaluating larger models in our experiments.
>
> Q2: How long would searching for a 2690 intra-layer parallelism candidates take in practice?
>
> A2: Let's take training GPT-3 with 175 billion parameters and a batch size of 512 as an example. In our experiments on distributed environment *EnvB*, enumerating each pipeline hyperparameter for single-layer-type model takes about 6.07 seconds on average. For a 512 batch size configuration, UOP will enumerate 241 different configurations. Therefore, it would take approximately 24 minutes (6.07s * 241 = 1462.87s) to seach for an optimal parallel strategy for 175B GPT-3. Additionally, due to our 30s constraint on each strategy searching time (please refer to Appendix D), the total strategy searching time would be bounded by 2 hours (30s * 241 = 7230s).
>
>
> Q3: Concerns on the MFU for BERT-Huge on EnvB. Are these baselines well-optimized?
>
> A3: We have calculated the MFU for BERT-Huge on *EnvA* and found that UniAP, Galvatron, and Alpa achieved an MFU of approximately 58.44%, 58.44%, and 55.10%, respectively. These results are slightly higher than the two references mentioned. Regarding the relatively low MFU on *EnvB*, we attribute this result to the significantly lower bandwidth between GPUs and nodes on *EnvB* than *EnvA*. Please refer to Figure 6 on pp.16 and discussions in Appendix E. Hence, we can find that both the baselines and UniAP are well-optimized. We will cite these references in our final version and discuss these findings.

---

### Author Response · Authors · 2023-11-20
**General responses**

We express our gratitude to the anonymous reviewers, ACs, and PCs for your careful consideration of our article and constructive comments. We believe that UniAP represents an innovative approach to solving the automatic parallelism problem. It addresses both inter- and intra-layer parallelism jointly within a larger strategy space and demonstrates the effectiveness of the approach on different clusters and models. We acknowledge the concerns raised by the reviewers and wish to address them by revising our paper and supplementing the experiments.

In response to general concerns about our main contributions (Reviwer pnV8 and Ye4S), we would like to clarify the following points:

1. UniAP isn't a cost model, but rather a framework for seeking a unified inter- and intra-layer parallelism strategy. For automatic parallelism frameworks like UniAP, it's essential to model the end-to-end (E2E) performance for model training to establish an objective function for minimization. In UniAP, we introduce an MIQP solution to jointly optimize a hybrid inter- and intra-layer parallelism strategy with globally minimal cost. Our experiments demonstrate that UniAP accelerates model training on four Transformer-like models by up to 1.7$\times$, highlighting the effectiveness of our approach.
2. Extending ILP to MIQP is a non-trivial task. While an intra-layer-only parallelism strategy aligns with the SPMD programming paradigm, integrating inter-layer parallelism requires unifying the non-SPMD programming paradigm with the SPMD one. Additionally, different styles of pipeline parallelism introduce additional complexity in abstracting the formulation while maintaining consistency. The resulting formulation is easy to read, but the efforts to simplify it are substantial.

The main modifications and additional experiments in response to other comments have been highlighted in blue in the paper and are briefly summarized as follows:

C1: Some of the details are hard to follow. (Reviewer pnV8)

R1: We have clarified ambiguous terms such as "communication efficiency" in our revision. Explanations for depicting the candidate solution have been added in Appendix F, which will aid in understanding the MIQP formulation.

C2: Experiments are not convincing. For example, all experiments lack error bars, evaluations on performance modeling and large scale clusters are not included. (Reviewer pnV8 and S34h)

R2: We have revised our experimental results to include error bars. Please refer to Table 1 on pp.8, Table 2 on pp.9, and Table 4 on pp. 18. Additionally, performance modeling results have been added in Appendix G.

Regarding larger scale experiments, we were unable to conduct evaluation or simulate results on larger scale clusters due to resource constraint in our lab which is from university. However, to the best of our knowledge, the scale of clusters (*EnvA*, *EnvB* and *EnvC*) in our experiment is almost the largest one for most universities. Meanwhile, many vision models have about 1B parameters. Therefore, optimizing parallel strategies for efficiently training a 1B model is also meaningful and our method provides a potential way for doing research on large language models for these universities. Furthermore, the scale of the cluster is not a strong assumption for UniAP. On the contrary, UniAP scales well based on our theoretical analysis in Appendix C and preliminary experimental results in Section 4.2. Hence, we acknowledge the limitation of not including results on larger scale clusters, but for a university lab, we have made our best effort to achieve a relatively promising result.

---

### Meta-Review · Area_Chair_QcQR · 2023-12-10

**Metareview:**

This paper was borderline in scores, and no reviewer showed particular enthusiasm about it. The more negative reviewer gave an extensive review detailing a number of specific issues and wasn't convinced by the authors' response. The most important issue concerns the scalability of the proposed method to larger models and number of GPUs (which is really very small in the experiments, 4 nodes and 16 GPUs at most). This is a critical shortcoming in a paper about parallel training of large neural nets. In an actual system implementation, where it is hard to quantify e.g. the costs of communication, computation, etc., a theoretical analysis of the speedup can be far off the real results; one really needs those experiments.

**Justification For Why Not Higher Score:**

See metareview

**Justification For Why Not Lower Score:**

N/A

---

### Decision · Program_Chairs · 2024-01-16

Reject